# *Lactiplantibacillus plantarum* 1201 Inhibits Intestinal Infection of *Salmonella enterica* subsp. *enterica* Serovar Typhimurium Strain ATCC 13311 in Mice with High-Fat Diet

**DOI:** 10.3390/foods11010085

**Published:** 2021-12-29

**Authors:** Zhongyue Ren, Lingling Peng, Shufang Chen, Yi Pu, Huihui Lv, Hua Wei, Cuixiang Wan

**Affiliations:** 1State Key Laboratory of Food Science and Technology, Nanchang University, Nanchang 330047, China; zhongyueR666@163.com (Z.R.); penglingling6@163.com (L.P.); shufangchen2021@163.com (S.C.); puyisunyyb@163.com (Y.P.); ncuspylvhuihui@163.com (H.L.); weihua@ncu.edu.cn (H.W.); 2School of Food Science and Technology, Nanchang University, Nanchang 330047, China; 3Sino-German Joint Research Institute, Nanchang University, Nanchang 330047, China

**Keywords:** *Lactiplantibacillus plantarum* 1201, *Salmonella* Typhimurium ATCC 13311, high-fat diet, intestinal infection, colonization, inflammation

## Abstract

*Salmonella* Typhimurium is widely distributed in food. It can colonise the gastrointestinal tract after ingestion, causing lamina propria edema, inflammatory cell infiltration, and mucosal epithelial decomposition. A high-fat diet (HFD) can induce an inflammatory response, but whether HFD can increase the infection level of *S.* Typhimurium is unknown. We established a model of *Salmonella enterica* subsp. *enterica* serovar Typhimurium strain ATCC 13311 ATCC 13311 infection in healthy adult mice with a maintenance diet (MD) or HFD to explore the effect of *Lactiplantibacillus plantarum* 1201 intervention on *S.* Typhimurium ATCC 13311 colonization and its protective effects on mice. HFD exacerbated the infection of *S.* Typhimurium ATCC 13311, while the intervention of *L. plantarum* 1201 effectively mitigated this process. *L. plantarum* 1201 can reduce the colonies of *S.* ATCC 13311 in the intestines and tissues; and reduce intestinal inflammation by down-regulating the level of TLR4/NF-κB pathway related proteins in serum and the expression of related inflammatory factors in the colon and jejunum. Since *L. plantarum* 1201 can inhibit the colonization of *S.* Typhimurium ATCC 13311 and relieve inflammation in HFD, current research may support the use of *L. plantarum* 1201 to prevent *S.* Typhimurium infection.

## 1. Introduction

With the continuous development of society, we are consuming ever more high-fat foods, especially those of animal origin, including beef [1], pork [2], poultry [3] and turkey [4]. *Salmonella* Typhimurium has been found on these foods, which can cause disease. According to statistics, more than 90% of bacterial-related food poisoning cases worldwide are caused by *Salmonella* and *Campylobacter* [5]. However, non-animal products, such as fresh vegetables and fruits, fruit juices and spices, are also associated with infection [6,7,8]. Because of this, is there any relationship between high-fat diet (HFD) and *S.* Typhimurium infection? *S.* Typhimurium can evade the host’s natural immune system, replicate in the host and cause disease [9]. Humans infected with *S.* Typhimurium show loss of appetite, vomiting and diarrhea, and may even develop sepsis [10]. At present, there is no way to control *S.* Typhimurium without side effects. Probiotic bacteria are reported to prevent the adherence, establishment and invasion of specific enteropathogenesis [11]. Several mechanisms have been proposed: contribution to mucosal barrier function, competitive exclusion, modulation of the immune response, coaggregation to pathogens, decreasing luminal pH via the production of lactic acid and secretion of specific compounds such as bacteriocins [12,13,14]. Therefore, we may start research from the direction of intestinal flora and colonization.

Studies have found that probiotics can colonise the gastrointestinal tract, supplement beneficial flora, maintain the balance of intestinal bacteria, and competitively inhibit the colonization of pathogenic bacteria. In addition, probiotics have an antagonistic effect on pathogenic bacteria in the intestinal tract and in food [15]. Some probiotics can produce beneficial metabolites, such as organic acids or hydrogen peroxide and natural antibiotics, forming an environment that inhibits or kills harmful bacteria. These acids can lower the pH value of the animal’s intestinal tract, effectively inhibit the growth of pathogenic bacteria, and create conditions for the growth of lactic acid bacteria. *Lactobacillus*, one of the most widely used probiotics, can form a biological barrier by regularly colonizing on mucous membranes, skin and other surfaces or between cells, thereby preventing the colonization of pathogenic microorganisms and inhibiting the survival of pathogenic bacteria. *Limosilactobacillus reuteri* can maintain the balance of pathogens and probiotics by neutralizing *Porphyromonas gingivalis*, thereby inhibiting inflammation and promoting wound healing [16]. *Lacticaseibacillus rhamnosus* can inhibit the colonization of *Vibrio parahaemolyticus* in the intestine [17]. *Lactobacillus gasseri* can inhibit the colonization of *Campylobacter jejuni* in the gastrointestinal tract [18]. After attaching to the cell surface, *Lactobacillus can* effectively control the reattachment of *Salmonella* bacteria to cells [19]. *L. plantarum* 1201 was isolated from Fuzhou preserved pickles. The antimicrobial activities experiment proved that *L. plantarum* 1201 has a better inhibitory ability against *Salmonella* Typhimurium which meant *L. plantarum* 1201 might inhibit the *S.* Typhimurium ATCC 13311 infections. Therefore, in this study, a mouse model of *S.* Typhimurium ATCC 13311 infection was constructed to investigate whether *L. plantarum* 1201 could inhibit *S.* Typhimurium ATCC 13311 infections, aiming to provide a new perspective for the treatment of *S.* Typhimurium infection.

## 2. Materials and Methods

### 2.1. Lactiplantibacillus Strains and Culture Conditions

*L. plantarum* 1201 was culture under anaerobic conditions at 37 °C in sterile deMan, Rogosa, Sharpe broth (Beijing Solarbio Science and Technology Co., Ltd., Beijing, China). Subsequently, cells were harvested, centrifuged at 5000× *g* for 5 min at 25 °C, and washed in sterile phosphate buffer saline (PBS). *L. plantarum* 1201 isolated from fermented acid beans is deposited in the China Center for Type Culture Collection with the collection number CCTCC M 2021050.

### 2.2. Animals and Experimental Design 

Eight-week-old specific pathogen free (SPF) female KM mice (Hunan Slack Jingda Experimental Animal Co., Ltd.) were assigned to cages, unloaded from the car in random order, and then randomly assigned into six groups. They were raised in the animal facility of Nanchang University, Jiangxi Province, under standard conditions with a light/dark cycle of 12 h. All mice were provided with ad libitum access to food and water. All experimental procedures are in accordance with the guidelines of the National Institutes of Health and were approved by the local animal health and use committee of Nanchang University. All pure feeds of the control group (MD group) and the experimental group were purchased from the China Nutrition Animal Feed High-Tech Co. Ltd. After 1 week of acclimatization in the sterile mouse room, the mice in the MD group, MD2 group and MD3 group (*n* = 8) were given a normal diet and normal drinking water, while the mice in the HFD group, HFD2 group and HFD3 group (*n* = 8) were maintained on an HFD (60% calories from fat) and normal drinking water. Then, the MD and HFD groups (*n* = 8) were intra-gastrically administered with 200 μL of 1 × PBS, and the MD2 and HFD2 groups (*n* = 8) were intra-gastrically administered with 100 μL of 1 × PBS and 100 μL of 1 × 10^8^ CFU/mL *S.* Typhimurium ATCC 13311, MD3 and HFD3 groups (*n* = 8) were intra-gastrically administered with 100 µL 1 × 10^8^ CFU/mL *L. plantarum* 1201 and 100 µL 1 × 10^8^ CFU/mL *S.* Typhimurium ATCC 13311. Mouse feces were collected at 3 h, 6 h, 24 h, 48 h, 72 h, and 96 h after gavage with strain. After 4 days of intra-gastric administration, the mice were euthanized with ether. The mouse serum, spleen, mesenteric lymph nodes (mLN), colon, cecum, jejunum and colon contents were collected in a sterile centrifuge tube, respectively, and stored at −80 °C for subsequent experiments.

### 2.3. Determination of Infection in Mice 

*S.* Typhimurium ATCC 13311 was incubated with 20 μg/mL Hoechst 33342 for 15 min [20]. All the samples including fecal, spleen, mLN, jejunum and colon contents were collected and weighed, then they were homogenized in PBS with grinding beads in a grinder. The homogenate was diluted gradually and spread on *Salmonella Shigella* (SS) agar selection medium [21]. After incubating for 24 h at 37 °C under aerobic conditions, we counted the number of colonies. The number of *S.* Typhimurium in the sample was calculated, and then the bacterial load of *S.* Typhimurium per gram of sample was calculated. The fluorescence intensity of the mouse fecal homogenate 6 h after gavage was observed under an inverted microscope.

### 2.4. Hematoxylin-Eosin Staining and Histopathological Damage Scores

Fresh colon tissues were collected and soaked in 10% formalin tissue fixative solution. After the alcohol was dehydrated, the tissue blocks were sequentially transparent, waxed, encased, sliced, and baked in turn. The conventional dewaxing process is to sequentially put paraffin sections into xylene I (10 min)—xylene II (10 min)—ethanol I (5 min)—ethanol II (5 min)—95% alcohol (3 min)—90% alcohol (3 min)—80% alcohol (2 min)—70% alcohol (2 min). Then, the slices were soaked in distilled water and washed for 2 min, and stained with hematoxylin and eosin (HE), anhydrous ethanol I 5 min, anhydrous ethanol II 5 min, xylene I in 5 min, and xylene II 5 min, then dried to a neutral gum sealing piece, and finally observed microscopy. Pathological score was determined by analysing four markers of inflammation: (1) submucosal oedema, (2) polymorphonuclear granulocyte infiltration into the lamina propria, (3) number of goblet cells harbouring mucus-filled vacuoles and (4) epithelial integrity. Pathological scores ranged from 0 to 13 arbitrary units (0–3, no or minimal inflammation; 4–6, slight inflammation; 7–9, moderate inflammation; 10–13, profound inflammation) [22,23].

### 2.5. Analysis of Inflammatory Levels in Intestine of the Mice 

High-quality RNA was isolated from frozen colon and jejunum tissue using the TaKaRa RNA extraction kit (Takara, Otsu, Japan), and then used for cDNA synthesis with a transcriptor cDNA kit (Takara, Otsu, Japan) according to the manufacturer’s protocol. The mRNA levels of the pro-inflammatory factors (IFN-γ, IL-1β, TNF-α, IL-6, IL-17A), anti-inflammatory factors (IL-10, IL-22, TGF-β) and NF-κB inflammation pathway-related genes (NF-κB, IκB-α) in the colon and jejunum tissue were measured using real-time PCR. Three-step PCR reaction procedure: 5 µL SYBR Green, 0.8 µL primer (10 μM), 1 µL cDNA (1000 ng/μL) plus ddH_2_O supplement 10 µL system. Reaction conditions were: preheat at 95 °C for 30 s, cycling stage: denaturation at 95 °C for 5 s; 59 °C annealing for 1 min; 72 °C extension for 30 s, 40 cycles, and then cooling to 65 °C for 5 s. The primer sequence information is shown in the Appendix A. 

### 2.6. Enzyme-Linked Immunosorbent Assay (ELISA) Detection of TLR4/NF-κB Inflammation Pathway in Serum of the Mice

The protein levels of TLR4, MyD88, IKKβ, IκB-α and NF-κB in mouse serum were determined by the double antibody sandwich method according to the steps of the ELISA kit (MEIMIAN, Jiangsu, China). After being left overnight at 4 °C, the freshly collected plasma was centrifuged at 3000 r/min for 10 min, and the serum was collected for testing.

### 2.7. S. Typhimurium Inhibition Assay

The agar diffusion assay was used to detect the inhibitory effect of *L. plantarum* 1201 on *S.* Typhimurium ATCC 13311. In short, 200 μL aliquot of *S.* Typhimurium ATCC 13311 (adjusted to 10^8^ CFU/mL, 10^7^ CFU/mL, 10^6^ CFU/mL) as an indicator microorganism was spread on LB agar. After the bacterial solution was dry, we put the Oxford cup on the LB plate. Then, 200 μL of the supernatant of *L. plantarum* 1201 was cultured overnight, and MRS broth with pH = 3.5 and fresh MRS broth with pH = 7 were added to Oxford cups (with an outer diameter of 6.8 ± 0.1 mm, an inner diameter of 6.0 ± 0.1 mm, and a height of 10.0 ± 0.1 mm). The plate was incubated at 37 °C for 36 h under aerobic conditions, and the diameter of the inhibition zone was measured. 

### 2.8. Statistical Analysis

Data analysis was carried out using GraphPad Prism 6 (GraphPad Software, La Jolla, CA, USA). One-way ANOVA was used for multiple comparisons, and all results are expressed as an average of S.E.M. The double-tailed unpaired Student *t* test was used for statistical evaluation. A *p* value of 0.05 was considered statistically significant.

## 3. Results

### 3.1. High-Fat Diet (HFD) Promotes S. Typhimurium ATCC 13311 Colonization while L. plantarum 1201 Inhibits It

We used SS selective agar medium to detect the colonization of *S.* Typhimurium. The levels of *S.* Typhimurium in the feces of the mice in the MD2, MD3, HFD2 and HFD3 groups that received *S.* Typhimurium were higher than those in the MD and HFD groups. The HFD2 group had the highest *S.* Typhimurium content. Interestingly, the relative colony number of the continuous high-fat diet group was higher than that of the normal diet group (Figure 1A–B). This is in agreement with a previous discovery [24]. Similarly, the relative colony number of *S.* Typhimurium in the organs of mice 96 h after intra-gastric administration also showed the same situation (Figure 1C). It is speculated that HFD can promote the colonization of *S.* Typhimurium, and *L. plantarum* 1201 has an inhibitory effect on the colonization of *S.* Typhimurium ATCC 13311.

### 3.2. L. plantarum 1201 Alleviates the Intestinal Injury Induced by S. Typhimurium ATCC 13311

Colonization of *S.* Typhimurium ATCC 13311 usually leads to intestinal pathophysiology. By comparing the sections of the colon of mice with hematoxylin-eosin staining and pathological scores, we observed that *S.* Typhimurium ATCC 13311 caused inflammation in the colon, and a high-fat diet could exacerbate this inflammatory response (Figure 2A,B). Compared with the MD group, the MD2 group mice that received a normal diet and received the *S.* Typhimurium ATCC 13311 gavage showed significant intestinal inflammation, while the HFD2 group that received a high-fat diet and gavaged *S.* Typhimurium ATCC 13311 showed more serious intestinal inflammation, with close arrangement and abnormal morphology of the intestinal villi and the number of abnormal goblet cells. However, in the MD3 and HFD3 groups treated with *L. plantarum* 1201, the number of abnormal goblet cells in mice decreased and the intestinal margin became clear, the intestinal villi were closely arranged, and the inflammation was relieved (Figure 2A,B). Therefore, it is speculated that HFD promoted the damage of *S.* Typhimurium ATCC 13311 to the intestinal tract, while *L. plantarum* 1201 reduced the intestinal inflammation induced by *S.* Typhimurium ATCC 13311.

### 3.3. L. plantarum 1201 Relieves Intestinal Inflammation through the TLR4/NF-κB Inflammatory Pathway

The levels of Toll-like receptor TLR4, myeloid differentiation factor MyD88, I-κB kinase IKK-β, nuclear transcription factor inhibitor IκB-α, and nuclear transcription factor NF-κB in serum were detected with an ELISA kit. At the same time, the mRNA expression levels of NF-κB, IκB-α in jejunum and cecum were detected. Compared with the MD group, the TLR4, MyD88, IKK-β, and NF-κB protein levels of the serum of the MD2 group and the HFD2 group were increased, and the HFD2 group showed a more significant increase in protein expression. However, in the MD3 and HFD3 groups that were given *L. plantarum* 1201, the protein expression level of TLR4, MyD88, IKK-β, NF-κB was down-regulated. In contrast, the protein expression level of IκB-α was down-regulated in the MD2 and HFD2 groups, but recovered in the MD3 and HFD3 groups (Figure 3B). The expression levels of NF-κB and IκB-α mRNA detected by qPCR also showed similar changes to those of protein levels (Figure 3A). It can be speculated that the colonization of *S.* Typhimurium ATCC 13311 caused inflammation by affecting the expression of related proteins in the TLR4/NF-kB inflammation pathway. HFD aggravated this change, and *L. plantarum* restored the inflammation pathway to inhibit inflammation.

### 3.4. L. plantarum 1201 Restores the Secretion of Intestinal Inflammatory Factors Caused by S. Typhimurium ATCC 13311 Colonization 

We detected the mRNA expression levels of interferon-γ, transforming growth factor TGF-β, tumor necrosis factor TNF-α, and interleukin IL-1β, IL-6, IL-10, IL-22, and IL-17A in jejunum and colon tissues. The results showed that the colonization of *S.* Typhimurium ATCC 13311 resulted in an increase in the mRNA expression levels of IL-1β, IL-6, IL-10, IL-22, IL-17A, TGF-β, TNF-α (Figure 4A,B). Therefore, we hypothesized that the colonization of *S.* Typhimurium ATCC 13311 would lead inflammation of the jejunum and colon. HFD aggravated this change and *L. plantarum* 1201 had different degrees of relief effects on it. Compared with the increased mRNA levels of inflammatory factors in the MD2 and HFD2 groups after ingesting *S.* Typhimurium ATCC 13311, under the intervention of *L. plantarum* 1201, the inflammatory mRNA expression levels in the MD3 and HFD3 groups were consistent with those in the MD and HFD groups, indicating that *L. plantarum* 1201 can inhibit intestinal inflammation caused by *S.* Typhimurium.

### 3.5. L. plantarum 1201 Inhibits S. Typhimurium ATCC 13311 Growth Activity

Compared with the MRS medium control with pH = 7, the supernatant of *L. plantarum* 1201 showed significant anti-*S.* Typhimurium ATCC 13311 activity, and had the best antibacterial effect on *S.* Typhimurium ATCC 13311 at 10^6^ CFU/mL (Figure 5). In order to verify whether *L. plantarum* inhibits the growth of *S.* Typhimurium ATCC 13311 by producing an acidic environment, a set of MRS medium controls with pH = 3 was added. Compared with the MRS medium with pH = 7, the antibacterial effect of the MRS medium with pH = 3 was increased, but it was significantly lower than the fermentation supernatant of *L. plantarum* 1201 (Figure 5B). These results indicate that *L. plantarum* 1201 can significantly inhibit the growth activity of *S.* Typhimurium ATCC 13311, and this inhibition is not only performed by acid production.

## 4. Discussion

At present, HFD has been proven to promote inflammation [25,26]. With HFD, the incidence of intestinal inflammatory diseases has gradually increased. Obese mice showed a special immune response during oral infection with *Salmonella* [27] and HFD can exacerbate inflammatory bowel disease [28]. The intake of high-fat meat protein diets resulted in the impairment of the colon barrier through mucus suppression, downregulation of tight junctions, and gut inflammation in mice [29]. In this study, HFD mice showed higher colonization of *S.* Typhimurium, more serious intestinal inflammation and more secretion of pro-inflammatory factors. These demonstrated that HFD can promote *S.* Typhimurium ATCC 13311 infection and cause more severe inflammation.

The colonization of *S.* Typhimurium ATCC 13311 affects the balance of human intestinal flora and induces inflammation. How to inhibit *S.* Typhimurium infection has become a key research direction of social concern. In previous studies, probiotics have been found to have an inhibitory effect on *S.* Typhimurium. *Bacillus* probiotics can inhibit the activity of *S.* Typhimurium [30]. *Bifidobacterium* can down-regulate the gene expression of *S.* Typhimurium pathogenicity islands 1 and 2 [31]. High-yielding *L. casei* with linoleic acid can limit the growth and activity of *S.* Typhimurium [32]. *L. plantarum* can reduce the levels of *S.* Typhimurium in the livers and spleens of mice [33]. However, the mechanism by which *L. plantarum* inhibits *S.* Typhimurium infection is still unclear. In this study, we discussed the inhibition of *S.* Typhimurium ATCC 13311 infections by *L. plantarum* 1201.

In vivo, the first direct contact of *Salmonella* with a host is the adhesion to the surface of epithelial cells, and then it colonises the gastrointestinal tract [34]. This event is a prerequisite for the subsequent steps in pathogenesis that lead to mucosal infection, systemic spread and disease. Inhibition of the invasion of *Salmonella* into epithelial cells is the first step in disease prevention [35]. Our animal experiment results show that *L. plantarum* 1201 can reduce the colonization of *S.* Typhimurium ATCC 13311 in mice. This protective effect is similar to previous research. Mixed probiotics can inhibit the colonization of *C. jejuni* in the intestine [36]. *L. casei* LC2W can inhibit the colonization of *E. coli* O157:H7 in vivo through an in vivo imaging system [37]. *B. clausii* and *L. reuteri* can inhibit the colonization of *C. difficile* in the human intestine [38]. In summary, our research found that HFD can promote the colonization of *S.* Typhimurium ATCC 13311 in the host, while the protective effect of *L. plantarum* 1201 is achieved by reducing it.

The TLR4/NF-κB pathway is a common inflammation pathway induced by Gram-negative bacteria. Toll-like receptors are the main receptors for the innate immune system to recognize pathogenic microorganisms and play an important role in the innate immune response [39]. TLR4 is an important receptor for the transmembrane signal transduction of pathogenic microorganisms. It is closely related to the body’s innate immunity against infection and plays an important role in the early recognition of invading pathogenic microorganisms by the immune system [40]. In the inflammatory response, NF-κB is the most critical transcriptional regulatory factor. It plays an important role in normal cell metabolism and immune response, and participates in the MyD88-dependent signaling pathway in the TLR4 signaling pathway [41]. Many studies have shown that *S.* Typhimurium can activate NF-κB, thereby causing inflammation [42,43,44]. It has also been found that probiotics can inhibit the NF-κB pathway activated by *S.* Typhimurium. *L. plantarum* JSA22 can inhibit the activation of NF-κB induced by *S.* Typhimurium and reduce the secretion of IL-8 [45]. Our conclusions are consistent with the literature [46]. In our study, when mice ingested a large amount of *S.* Typhimurium ATCC 13311, TLR4 was activated and bound with MyD88, which, in turn causes the phosphorylation of serine residues at specific sites of IκB-α. Then, IκB-α is ubiquitinated and degraded, while NF-κB is released and transferred to the nucleus, inducing the production of inflammatory factors and promoting intestinal inflammation. HFD exacerbated this inflammatory response. However, the intervention of *L. plantarum* 1201 inhibited the colonization of *S.* Typhimurium, thereby inhibiting the activation of this inflammatory pathway. 

Previous studies have shown that after TLR4 activates the TLR4/NF-κB signaling pathway, it will cause an immune response against microbial infections and release various inflammatory factors, such as TNF-α [47]. Inflammatory factors play an important role in the infection process of *S.* Typhimurium. In the case of inflammation, necrosis, or immune cells stimulated by tumor cell antigens, the secretion of IL-6 increases, which reflects an increase in bacterial colonization in tissues [48,49]. In contrast, IFN-γ and IL-17A have been shown to help the host resist *S.* Typhimurium. The increased secretion of these two cytokines reflects the enhanced protective immune response regulated by the microflora in mice [50,51]. CD4+ T cells can generate the transforming growth factor TGF-β, and TGF-β can induce the differentiation of Th17 cells or Treg cells to produce the inflammatory factor IL-17. Therefore, TGF- β has a role in inflammatory response and immune regulation [52]. TNF-α is a pro-inflammatory cytokine produced by specific cells under inflammatory stimulation and has a variety of biological activities [53]. It can expose the nuclear localization sequence of NF-κB by mediating the phosphorylation and ubiquitination of IκB. NF-κB then translocates into the nucleus and binds to the NF-κB site in the nucleus to initiate gene transcription and induce the massive release of cytokines including TNF-α. The released cytokines can further activate NF-κB, so that the positive feedback makes the inflammatory response progressively amplified [54]. IL-10 is another key cytokine for *S.* Typhimurium infection. The increase in IL-10 will improve the body’s immune tolerance, prevent *S.* Typhimurium from being eliminated, and trigger the spread of the system [55]. Interleukin-1β is an important mediator of the inflammatory response, and is involved in a variety of cellular activities, including cell proliferation, differentiation, and apoptosis [56]. IL-22 can effectively regulate inflammation, especially in inflammation caused by pathogenic bacteria and prevent the colonization and spread of *S.* Typhimurium in the intestine by maintaining the integrity of the epithelial barrier, thereby limiting bacterial growth and reducing inflammation [57,58]. 

From the qPCR results, we can see that *S.* Typhimurium colonization can promote the mRNA expression of inflammation-related cytokines, enhance the tissue’s inflammatory response and damage the body, which can promote the occurrence of intestinal inflammation. The situation in combination with an HFD is more pronounced. However, *L. plantarum* 1201 has significant alleviating effects on the inflammation and body damage caused by *S.* Typhimurium infection. 

## 5. Conclusions

*S.* Typhimurium ATCC 13311 can activate the TLR4/NF-κB inflammatory pathway after colonization, promote the abnormal secretion of inflammatory factors, and cause intestinal inflammation. HFD can facilitate this process. *L. plantarum* 1201 can inhibit *S.* Typhimurium ATCC 13311 infections by inhibiting its colonization and inflammation pathways (Figure 6). It is expected that the *L. plantarum* 1201 can be used as a new target to protect human health from infection of *S*. Typhimurium through diet.

## Figures and Tables

**Figure 1 foods-11-00085-f001:**
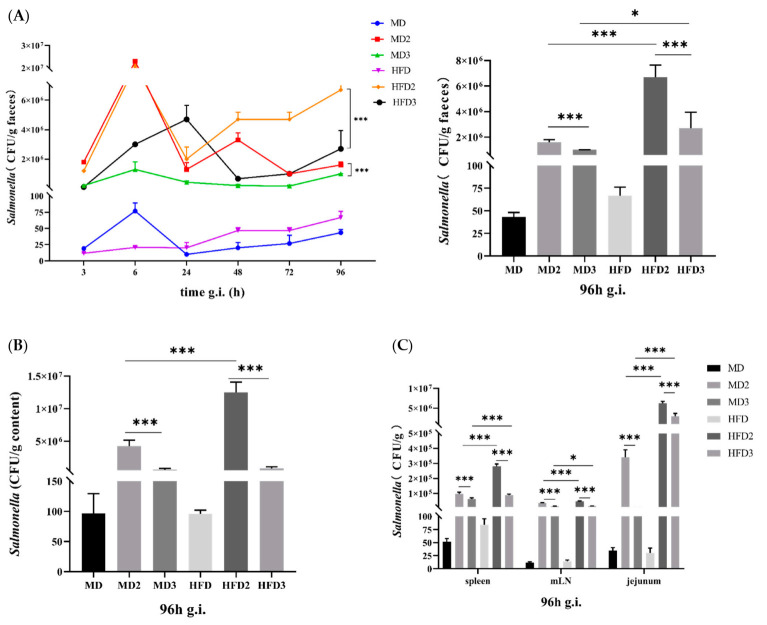
High-fat diet (HFD) promotes *S.* Typhimurium ATCC 13311 colonization while *L. plantarum* 1201 inhibits it. (**A**) statistical graph of *S.* Typhimurium content in mouse feces after gavage and the content of *S.* Typhimurium in mouse feces after intra-gastric administration for 96 h; (**B**) statistical graph of the content of *S.* Typhimurium in the colon content of mice after intra-gastric administration for 96 h; (**C**) statistical graph of the content of *S.* Typhimurium in mouse organs after intra-gastric administration for 96 h. * *p* < 0.05; *** *p* < 0.001; paired two-tailed t-test. MD: normal diet; MD2: normal diet and 1 × 10^8^ CFU/mL *S.* Typhimurium ATCC 13311; MD3: normal diet, 1 × 10^8^ CFU/mL *S.* Typhimurium ATCC 13311 and 1 × 10^8^ CFU/mL *L. plantarum* 1201; HFD: high-fat diet (HFD); HFD2: HFD and 1 × 10^8^ CFU/mL *S.* Typhimurium ATCC 13311; HFD3: HFD, 1 × 10^8^ CFU/mL *S.* Typhimurium ATCC 13311 and 1 × 10^8^ CFU/mL *L. plantarum* 1201; mLN, mesenteric lymph node; g.i., gavage infection.

**Figure 2 foods-11-00085-f002:**
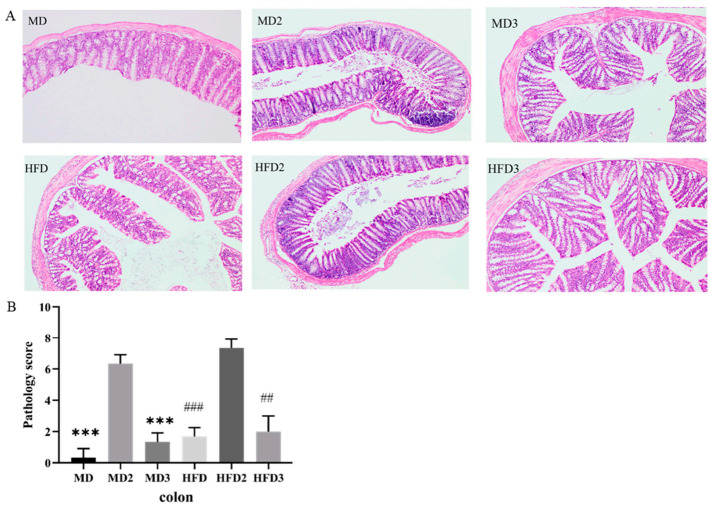
*L. plantarum* 1201 alleviates the intestinal injury induced by *S.* Typhimurium ATCC 13311. (**A**) hematoxylin-eosin staining (HE) pathological section of mouse colon tissue; (**B**) Colon histopathological score chart. *** *p* < 0.001; ## *p* < 0.01; ### *p* < 0.001; * is compared with MD2, # is compared with HFD2; paired two-tailed *t*-test. MD: normal diet; MD2: normal diet and 1 × 10^8^ CFU/mL *S.* Typhimurium ATCC 13311; MD3: normal diet, 1 × 10^8^ CFU/mL *S.* Typhimurium ATCC 13311 and 1 × 10^8^ CFU/mL *L. plantarum* 1201; HFD: HFD; HFD2: HFD and 1 × 10^8^ CFU/mL *S.* Typhimurium ATCC 13311; HFD3: HFD, 1 × 10^8^ CFU/mL *S.* Typhimurium ATCC 13311 and 1 × 10^8^ CFU/mL *L. plantarum* 1201.

**Figure 3 foods-11-00085-f003:**
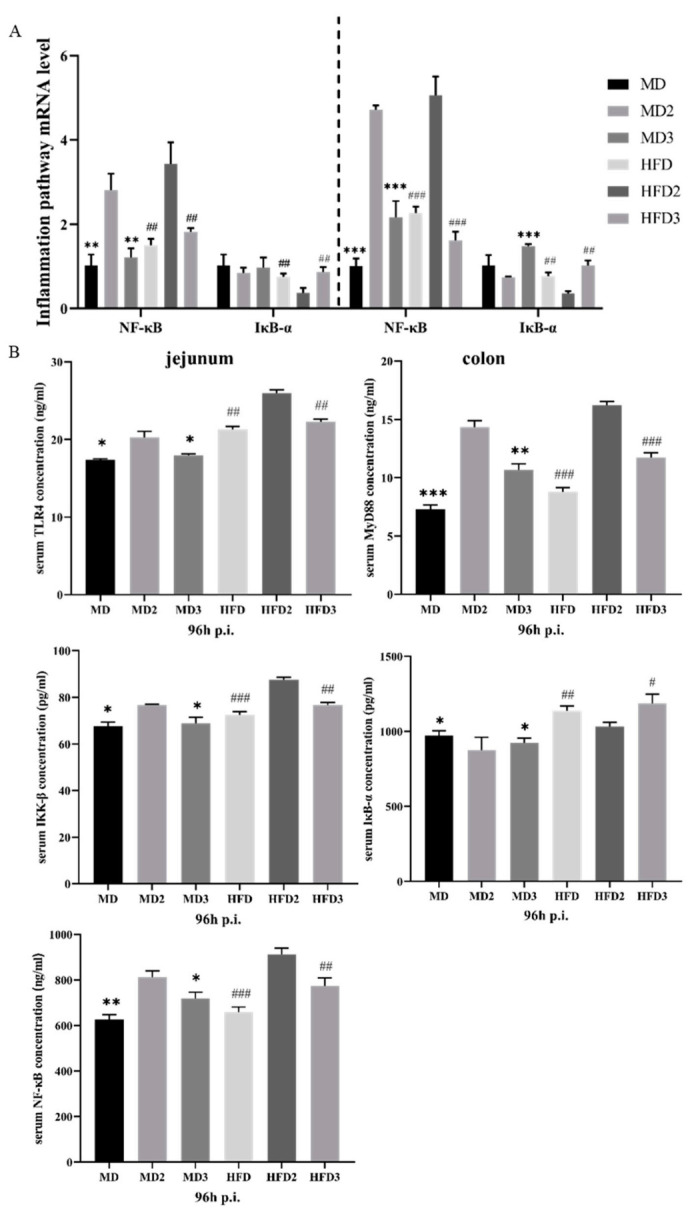
*L. plantarum* 1201 relieves intestinal inflammation through the TLR4/NF-κB inflammatory pathway. (**A**) mRNA expression of NF-kB inflammation pathway in colon and jejunum of mice in different groups; (**B**) concentration of TLR4/NF-κB pathway protein in mouse serum. * *p* < 0.05; ** *p* < 0.01; *** *p* < 0.001; # *p* < 0.05; ## *p* < 0.01; ### *p* < 0.001; * is compared with MD2, # is compared with HFD2; paired two-tailed t-test. MD: normal diet; MD2: normal diet and 1 × 10^8^ CFU/mL *S.* Typhimurium ATCC 13311; MD3: normal diet, 1 × 10^8^ CFU/mL *S.* Typhimurium ATCC 13311 and 1 × 10^8^ CFU/mL *L. plantarum* 1201; HFD: HFD; HFD2: HFD and 1 × 10^8^ CFU/mL *S.* Typhimurium ATCC 13311; HFD3: HFD, 1 × 10^8^ CFU/mL *S.* Typhimurium ATCC 13311 and 1 × 10^8^ CFU/mL *L. plantarum* 1201; p.i., post infection.

**Figure 4 foods-11-00085-f004:**
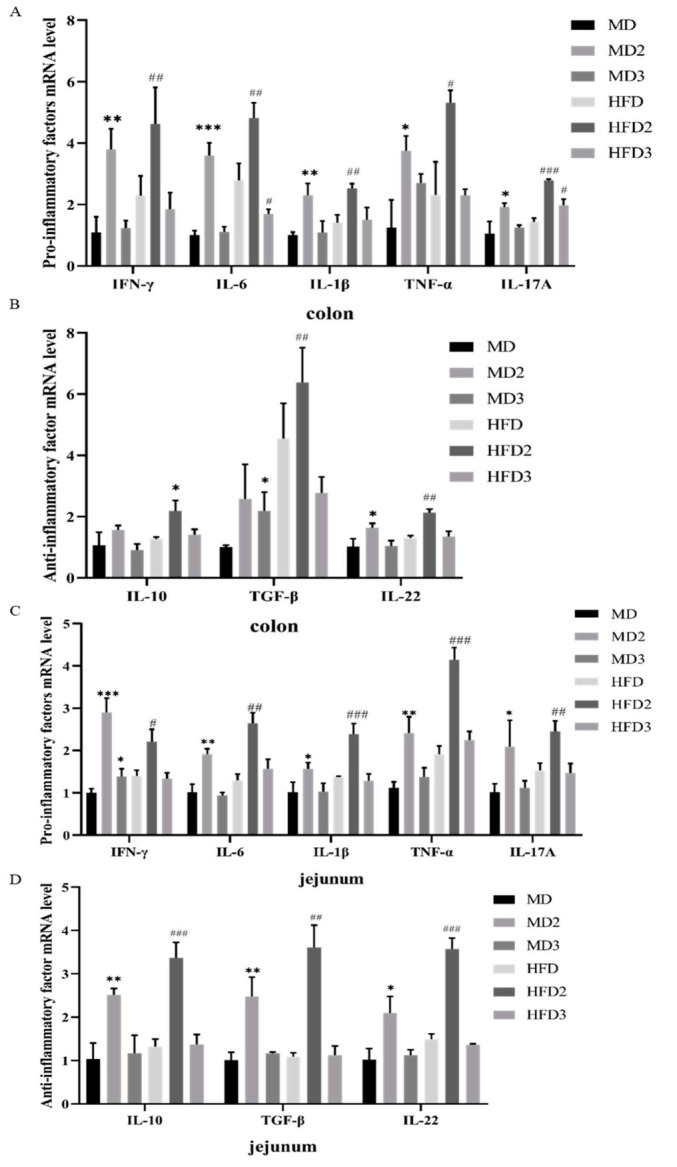
*L. plantarum* 1201 recovers the expression disorder of inflammatory factors caused by *S.* Typhimurium ATCC 13311. (**A**) mRNA expression levels of pro-inflammatory factors in mouse colon; (**B**) mRNA expression levels of anti-inflammatory factor in mouse colon; (**C**) mRNA expression levels of pro-inflammatory factors in mouse jejunum; (**D**) mRNA expression levels of anti-inflammatory factors in mouse jejunum. * *p* < 0.05; ** *p* < 0.01; *** *p* < 0.001; # *p* < 0.05; ## *p* < 0.01; ### *p* < 0.001; * is compared with MD, # is compared with HFD; paired two-tailed t-test. MD: normal diet; MD2: normal diet and 1 × 10^8^ CFU/mL *S.* Typhimurium ATCC 13311; MD3: normal diet, 1 × 10^8^ CFU/mL *S.* Typhimurium ATCC 13311 and 1 × 10^8^ CFU/mL *L. plantarum* 1201; HFD: HFD; HFD2: HFD and 1 × 10^8^ CFU/mL *S.* Typhimurium ATCC 13311; HFD3: HFD, 1 × 10^8^ CFU/mL *S.* Typhimurium ATCC 13311 and 1 × 10^8^ CFU/mL *L. plantarum* 1201.

**Figure 5 foods-11-00085-f005:**
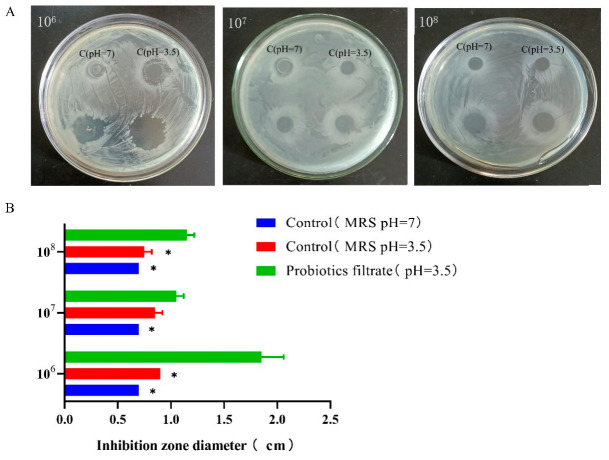
*L. plantarum* 1201 inhibits the growth of *S.* Typhimurium ATCC 13311. (**A**) Intuitive diagram of inhibition zone; (**B**) Bar graph of inhibition zone. * *p* < 0.05; paired two-tailed *t*-test.

**Figure 6 foods-11-00085-f006:**
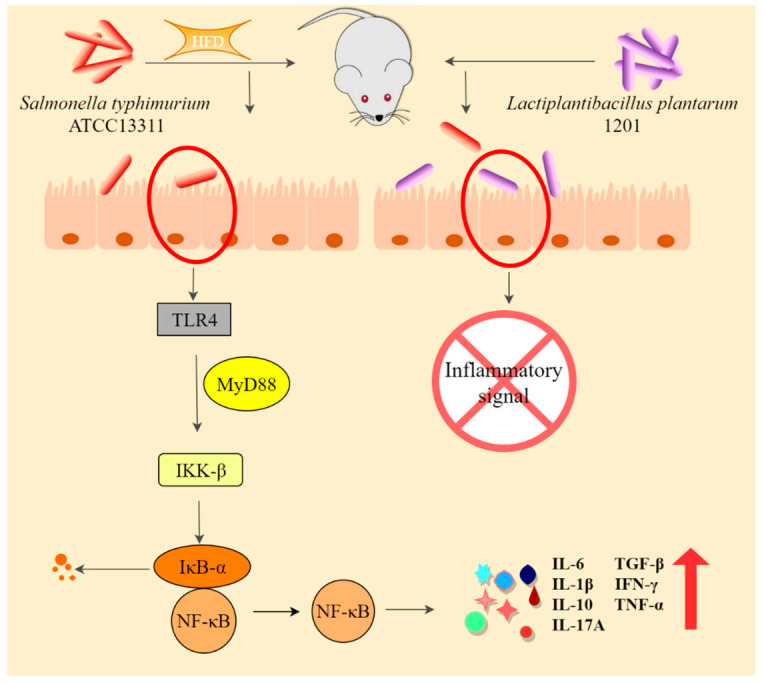
HFD exacerbates the intestinal infection of *S.* Typhimurium ATCC 13311, while *L. plantarum* 1201 inhibits it.

## Data Availability

The data presented in this study are available in the article and its Appendix A.

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
