# Peer review of "Lactiplantibacillus plantarum* 1201 Inhibits Intestinal Infection of *Salmonella enterica* subsp. *enterica* Serovar Typhimurium Strain ATCC 13311 in Mice with High-Fat Diet"

_foods, 2021, doi:10.3390/foods11010085_

Round 1

Reviewer 1 Report

The authors study the effect of Lactiplantibacillus plantarum 1201 on S. typhimurium ATCC13311 infection and a high-fat diet. The manuscript is interesting, and the results can be useful.

However, some minor aspects are itemized below with some questions to be solved in order to improve the manuscript:

- 2. Materials and Methods: The type of fat provided in the diet has not been characterized. Perhaps authors could explain if the type of fat would influence the results.

- 4. Discussion: both pro-inflamatory and anti-inflamatory factors are always diminished due to L. plantarum effects (Fig.4). The authors should explain in more depth the heath advantages and  disadvantages of this fact.

- References: authors must correct the style of all references, and standardise it: in some article titles, all words start with a capital letter in other titles all words are written in lower case; the names of magazines are sometimes in abbreviations, sometimes in full names; sometimes doi code are added, sometimes not, etc.

Reviewer 2 Report

The authors employed an animal model to investigate the role of Lactiplantibacillus plantarum 1201 in attenuating Salmonella of the serotype Typhimurium ATCC13311 infection. In the model, two groups of mice were used - one with a maintenance diet (MD) which was the control group and the second with a high-Fat diet (HFD), since the state of obesity can contribute to inflammation. The authors found that L. plantarum 1201 can inhibit colonization of S. Typhimurium ATCC13311 and alleviate inflammation-21 in HFD. Since the text is neither clear nor concise, the manuscript as a whole would benefit from an English revision. Furthermore, there are many questions that need to be clarified. Here are some comments and questions:

- The nomenclature of the Lactobacillus genus has undergone a recent change. In the title of the manuscript, the authors correctly use the new designation for the species. Therefore, they should also use it in the text.

- Lines 30 – 31: please, authors should confirm whether this information is correct. The frequency of one third of all cases of foodborne infection associated only with the Typhimurium serotype is very high. Additionally, I inform that the reference used in the citation is related to a study with the Newport serotype, which is a non-nontyphoidal serotype;

- Line 35 – 36: reference 10 concerns the protective action of Lactobacillus kefir against Salmonella enterica of the Enteritidis serotype. The study does not describe the symptomatology of S. Typhimurium infection;- Lines 37 – 38: It is written that “Probiotic has been proved to possess the function of resisting the invasion of pathogenic bacteria”. In fact, probiotics can provide the host with resistance against invasion by pathogenic bacteria.

- Although SS agar is selective, how can the authors ensure that only Salmonella grew in stool samples?

- Results, Fig. 1: With the exception of the HFD2 group, the difference in fluorescence intensity in the other groups is not clear. Furthermore, the caption of the figure should have the definition of each group so that the reader does not have to return the methodology to identify each one. This figure is not clear – in item 2.2 of the methodology it is indicated that the MD and HFD groups did not receive Salmonella Typhimurium. So what does the Salmonella CFU count mean for these groups in the graphs? Another issue concerns the comparative analysis between groups. The caption indicates that the comparisons were with MD and HFD (groups that, according to the methodology, did not receive S. Typhimurium), but what about comparisons between MD2 and HFD2 (group with Salmonella) with MD3 and HFD3 (groups with Salmonella and Lactiplantibacillus plantarum)?

- Line 215 – 216: the sentence “L. plantarum 1201 had different degrees of relief effects on it” is too vague. Please be more specific or explain a little more;

Figure 5: Figure 5A has poor quality. Furthermore, there is an inconsistency between the figure and what is said in lines 224 -226, where it is said that the supernatant of L. plantarum 1201 showed significant anti-S. typhimurium ATCC13311 activity the best antibacterial effect on S. typhimurium ATCC13311 at 108 CFU/mL. However, it can be seen that the highest activity was at 106CFU/mL (5B);

- line 259: It's not clear what the authors meant with “the number of adhesions directly reflects the infection situation”.

Minor comments:

- Typhimurium is a serotype and therefore should not be italics;

- Lines 55 and 56: Lactobacillus and Salmonella should be in italics;

- Line 85: 72h, and 96h;

- Line 123: 59 oC annealing (not return);

Round 2

Reviewer 2 Report

The authors accepted the suggestions and the manuscript improved considerably. However, there are still few adjustments to be made in the text, particularly in taxonomy. So, here are other suggestions:

1) In the title and the first time you quote the Salmonella strain used in the text, use Salmonella enterica subsp. enterica serovar Typhimurium strain ATCC 13311. Afterwards the authors can abbreviate the genus, ie: S. Typhimurium ATCC 13311 (only the letter S in italics and the letter T in capitals);

2) Lines 57 – 66: When I called the authors' attention to the change in the taxonomy of Lactobacillus I should have mentioned that the other species were designated differently (see Zheng et al., Int. J. Syst. Evol . Microbiol. 2020;70:2782-2858, DOI 10.1099/ijsem.0.004107). Therefore, correcting the text:

Lactobacillus, one of the most widely used probiotics, can form a biological barrier by regularly colonizing mucous membranes, skin and other surfaces or between cells, thereby preventing the colonization of pathogenic microorganisms and inhibiting the survival of pathogenic bacteria. Limosilactobacillus reuteri can maintain the balance of pathogens and probiotics by neutralizing Porphyromonas gingivalis, thereby inhibiting inflammation and promoting wound healing[16]. Lacticaseibacillus rhamnosus can inhibit the colonization of Vibrio parahaemolyticus in the intestine[17]. Lactobacillus gasseri can inhibit the colonization of Campylobacter jejuni in the gastrointestinal tract[18].

3) Before citing the objective of the study, the authors should describe the strain Lactiplantibacillus plantarum 1201 (its origin and characteristics that led to its selection for the study, etc.)

4) Line 294: Bifidobacterium is misspelled
